# Study of the Materials and Techniques of a Rare Papier-Mâché Mushroom Model Crafted in H. Arnoldi Factory

**DOI:** 10.3390/molecules28031062

**Published:** 2023-01-20

**Authors:** Maria J. Melo, Ana Freitas, Cristiana Vieira, Márcia Vilarigues, Márcia Vieira, Paula Nabais, Sílvia Sequeira, Mónica Lourenço, Gabriel Oliveira, Ana Rita Correia

**Affiliations:** 1LAQV-REQUIMTE and Department of Conservation and Restoration, NOVA School of Sciences and Technology of NOVA University Lisbon, 2829-516 Monte da Caparica, Portugal; 2Gestão de Documentação e Informação Universidade do Porto Digital (GDI-UPDigital), Praça Gomes Teixeira, 4099-002 Porto, Portugal; 3Museu de História Natural e da Ciência da Universidade do Porto, PO Herbarium, Praça Gomes Teixeira, 4099-002 Porto, Portugal; 4VICARTE and Department of Conservation and Restoration, NOVA School of Sciences and Technology, NOVA University of Lisbon, 2829-516 Caparica, Portugal

**Keywords:** cultural heritage, spectroscopy techniques, conservation, PO Herbarium, university museum collections

## Abstract

The Natural History and Science Museum of the University of Porto houses a collection of 45 models of fungi in papier-mâché from the 19th-century, which were used at the university until 2015 as didactic models. For the first time, the materials and techniques used in the production of a *Boletus edulis* model were studied (vernacular name: cep, porcini). These sculptures, made to life-size scale, are painted in colors similar to those of the represented species (white, brown, and light brown). They are fixed to a rectangular base, which is painted black, and to which moss has been pasted. To fully characterize each color, at the molecular level, a multi-analytical approach was used, combining energy-dispersive x-ray fluorescence spectroscopy (micro-XRF) with fingerprinting techniques of Raman microscopy (microRaman and handheld Raman) spectroscopy and microFourier transform infrared spectroscopy (microFTIR). The papier-mâché was prepared with a groundwood paper to which kaolin and a quartz-based material have been added to reinforce the structure. Raman microscopy also identified carbon black in it, which is possibly responsible for its grey color. The white color was unequivocally identified as lithopone by microRaman. This white paint was prepared in a proteinaceous tempera, with calcium carbonate having been identified as filler (by microFTIR). In the brown color, iron was identified by microXRF, pointing to the use of ocher, which was not possible to identify by microRaman and microFTIR. Regarding the black rectangular base, the moss was fixed using a collagen-based glue. The binding medium in this black is possibly a mixture of drying oil and protein. Again, XRF detected iron as the main element, but it was not possible to acquire a Raman spectrum due to the high fluorescence of the binder/varnish. Others, such as the writing inks, will also be discussed. The colors identified are in line with the best materials available for use by artists of that time. This new knowledge is fundamental to informing the choice of the best conservation strategies for the preservation of these extraordinary models.

## 1. Introduction

### 1.1. Didactic Arnoldi’s Collection of Mushroom Models at the University of Porto Herbarium

#### 1.1.1. Rationale for the Study

The Fungi didactic model collection from the Porto Herbarium of the Natural History and Science Museum of the University of Porto (MHNC-UP) includes a set of 45 scale models of the reproductive organs of fungi (mushrooms) painted in natural colors to teach the characteristics and differences between edible and poisonous species. These were crafted by Heinrich Johannes Arnoldi (11 July 1813–28 December 1882) [1]. Though no receipts or direct mentions of acquisitions of the University of Porto Herbarium (PO) Arnoldi’s mushroom model collection have been traced in the account books of the Polytechnic Academy of Porto (the predecessor of the University of Porto, 1837–1911). When the regulation for the botanical garden’s teaching building was published, one of the functions of the first official was to maintain an updated inventory of all instruments including “models” [2]. The University and Arnoldi stamps attached to the extant models suggest that this mushroom model collection was acquired in the late 1800s, so they were probably purchased through bookstores that sold international botanical publications and didactic materials such as the ones advertised either by H. Arnoldi or other European *comptoirs*.

This collection of mushroom models had been used from the late 19th-century until 2015 as a didactic resource in classes, withstanding manipulation and classroom environment for more than a century. In 2015 they were integrated on the Natural History and Science Museum of the University of Porto collections and housed at the Herbarium reserve. Despite their perseverance, the Herbarium curator (C. Vieira) and the University paper conservator (A. Freitas) noticed the fragile condition of these models and decided to undergo a thorough characterization of the collection that would lead to their conservation and valuation. More about the PO Herbarium and its collections is below, in “Materials and Methods” [3].

At the beginning of this work, we could not find much information published on Arnoldi’s fungi models or the existing collections throughout the world (much more exists about Arnoldi’s fruit models). The Slovenian Museum of Natural History (Prirodoslovni muzej Slovenije), located in the Slovenian capital, Ljubljana, has in its collection 444 Arnoldi models of mushrooms with 254 different species, which were acquired before 1885 [4]. In order to gather information on Arnoldi’s mushroom collections throughout the world we performed an informal census through messages to the Global Conservation Forum (ConsDistList), the American Institute for Conservation (AIC), and the Foundation For Advancement In Conservation and, after consulting several sources of information such as specific publications [4,5], and curators of several institutions and online web pages, we found information on Arnoldi models at the Nature Museum of Olten (Naturmuseum Olten), Switzerland (296 models); Coburg Natural History Museum (Naturkundemuseum Coburg), Germany (73 models); Hannover State Museum (Landesmuseum Hannover), Germany (40 models); Santos Botanical Museum (Santos Museum of Economic Botany), Australia (210 models); and the University of Vienna, Austria (31 models).

The Arnoldi mushroom collection kept at the University of Porto, Portugal, is possibly a special edition and, it is interesting to study, characterize and preserve these 45 mushroom models, since we believe it is, so far, the only extant collection in Portugal. In the late 1800s, Julio Henriques (1838–1928), a professor at Coimbra University, Portugal, reported the acquisition of models from the H. Arnoldi company for their botanical museum and also for teaching. Nevertheless, these Coimbra mushroom models could not be found recently in the Coimbra collections.

#### 1.1.2. Arnoldi Manufacture

Heinrich Johannes Arnoldi (11 July 1813–28 December 1882) was a member of the Thüringen Pomologische Gesellschaft and the Thüringer Gartenbau Verein, and later co-owner of the Arnoldi family’s porcelain factory [5]. In 1865, the factory was renamed the “Fabrik künstlicher Früchte und Pilze”, as Arnoldi manufactured fruit models (“Arnoldisches Obstkabinett”) made of porcelain to illustrate the pomological richness of the region [1]. 

Arnoldi then began modelling in papier-mâché, making models easier to transport without breaking, producing fruit and mushroom models and selling them via a catalogue (*Arnoldische Pilzsammlung*). In 1894, Arnoldi issued a price list for his Gotha factory’s lifelike artificial fruits and mushrooms [1]. The mushroom models could be sold individually or in consignments. These collections were generally divided into a series of edible, inedible, or even toxic mushrooms. In the catalogue, the different models were described, and the customer could choose some of them or subscribe to receive the complete collection (420 species of mushrooms, distributed in 35 shipments in total, with 12 mushrooms in total available in 1894/95). These models came with a description, a fitting plate, and a wooden box, where they could be fixed for exhibition.

In the catalogue Arnoldi states “(…) *consisting of true-to-life, plastic reproductions with accompanying scientific descriptions, this is the oldest of such widely recognized undertakings and was first published in 1871 with the friendly assistance of Dr W. Gonnermann in Coburg, Professor O. Burbach in Gotha and other outstanding mycologists who were known as the first authorities* (…)”. Arnoldi also states that the mushroom collection received numerous medals and awards and that the growing use of the almost universally abundant edible mushrooms as a valuable and cheap food justifies the aim of this collection as a means for the dissemination of mycology in general and in economic terms. He claims that the specific objectives are to distinguish the edible species from the poisonous and non-poisonous ones and thus to eliminate the well-founded fear of poisoning, having as the highest priority the pedagogical teaching in all schools and institutions, and also to serve the day-to-day life of families. He also asks experts and friends to inform him of the mushrooms not yet included in the assortment, their scientific names and other explanations, so that the collection could expand even more [1].

Like its contemporary botanical models, the series of models from Arnoldi’s mushroom collections belonging to the MHNC-UP are distinctive and very detailed: in each realistic-scaled model (replicas), one or more stages of development of the fructifying bodies (mushrooms) of the species are fixed on a quadrangular black basis in which, at the foot of the stem, is added natural moss [1]. Both the upper surface and bottom surface of the base of the models have labels with the identification of the species in Latin, and the bottom label includes a description of the mushroom, in German.

Although the Arnoldi family sold the company as late as 1907, their fungi model production had been already taken over in 1890 by the German firm SOMSO^®®^ Modelle [4]. 

### 1.2. The Materials and Techniques Used in the Production of 19th-Century Didactic Models in Papier-Mâché

Around the end of the 19th century, many realistic mushroom replicas or model ma-nufacturers were known. Papier-mâché or other predecessor methods (including wax and resins) were seen in mushroom model collections produced by names such as James Sowerby, Leopold Trattinnick, Victor Dürfeld Nachf, Luigi Calamain, André-Pierre Pinson and Jean-Baptiste Barla, as well as the well-known Louis Auzoux [6], among others.

“Papier-mâché” is a term that has been applied to three-dimensional (3D) objects of recycled paper fiber, whether layered in sheet form with an adhesive or cast as beaten pulp [7]. It was invented in East Asia around 200 AD. and brought to Europe in the 18th century [8]. In Asia, they commonly used macerated and shredded paper mixed with copal varnishes (resins obtained from some tropical trees), vegetable gums, and other fibrous materials (e.g., rags) to give more resistance to the final object. The paper and fibers were usually softened in boiling water, squeezed and dried to obtain a workable mixture, and then the remaining components were added. The thickness of the final material could vary depending on the production method (with more or less water): it could be air-dried or baked in flat plates, or wood or hard clay molds with the shape of the intended object could be used [7]. In the case of molds, one could also create hollow objects [9]. Papier-mâché proved to be an alternative to wax, wood and porcelain, and quick large-scale production was possible, achieving objects resistant to the daily handling of teachers and students, to different climates and to transportation [10].

According to Pungaršek and Piltaver [4], the production of Arnoldi’s mushroom models began with the creation of a plaster mold, which was later filled with a mixture of papier-mâché and plaster [1] (p. 16). Then the two halves of the mold were joined and, after drying, the model was covered with plaster and left to dry. It would then be hand-painted to look as real as possible. A metallic or wooden rod was inserted inside the mushrooms to allow them to be attached to the wooden base. In special editions, real moss was glued in the base after it has been painted black and varnished [1].

Arnoldi’s mushroom models were described as being made with a mixture of papier-mâché and plaster, as were the many fruits in 19th-century pomological cabinets [4]. However, the composition varied according to recipes and factories, and because there is neither research related to the materials used by Heinrich Arnoldi & Co., nor studies and analyses carried out on the models, the specific composition is yet unknown. The material of the pictorial layer is also unknown. To the best of our knowledge, there are some publications on the conservation/restoration of papier-mâché models, for instance, Infrared spectroscopy studies of anatomical Auzoux and Brendel’s models [11]. However, no publications on the analysis of the color of Arnoldi didactic models of fungi are known. There is a recent publication on papier-mâché models for plants by Mayoni [10]. In this research, some samples were analyzed by SEM-EDS and the following stratigraphy was proposed for color construction: a first ground layer based on calcium carbonate (although SEM-EDS analyses identifies also the element Al and Si); a second white layer in which Zn was detected as “very pure”; a third layer in which both Zn and Pb were identified [10]. Complex infrared spectra of the binding media are also depicted in Figures 6 and 8 in this article; a protein is present in one of the samples and for the others, a resin of vegetal origin is proposed [11]. We will also use as a reference the only known publication approaching the pigments used in a Portuguese sculpture in papier-mâché representing Saint Anthony, probably produced in the 19th-century [7]. In this religious sculpture, pigments were analyzed by microXRF and microRaman. Pink colors were prepared admixing vermillion and a zinc-based pigment and the “original brown color consisted of iron and manganese-based pigment, likely umber (iron and manganese oxides)” [7].

### 1.3. An In-Depth Study of the Materials of the Boletus edulis Didactic Model Using a Multi-Analytical Approach

After a thorough examination of the several components of the model with naked eye and under the microscope, we began the molecular characterization of the colors and techniques used. Our analysis combines elemental (microXRF) techniques with spectroscopic techniques such as Raman microscopy (microRaman) and Fourier Transform Infrared spectroscopy (microFTIR), which are powerful complementary spectroscopic techniques for the characterization of heritage materials. Raman and infrared spectroscopy reveal a “molecular fingerprint”; if a single compound is present, it is possible to unequivocally characterize it [12]. Complexity arises because we are usually faced with aged mixtures of compounds. For this reason, assignments are made with our database of colorants, binders, and color paints [12,13]. For more details, please see the Materials and Methods section.

## 2. Results and Discussion

### 2.1. Visual Description and Condition of the Boletus edulis Didactic Model

After a general visual evaluation of the condition of the 45 models of Arnoldi’s mushroom collection housed at PO Herbarium, we decided to choose one of the most complex models (in terms of colors, pieces’ detachment possibility, and number of mushrooms attached to the base), *Boletus edulis*, Bull. (hereinafter, *Boletus edulis* or *B. edulis*), perform a thorough examination of the several components of the model with the naked eye and under the microscope, and begin a molecular characterization of the colors and techniques used. For more details, please see the Materials and Methods section.

This didactic model corresponds to one of the 45 Arnoldi models existing in the MHNC-UP collection. In Figure 1 we present a photo of the *Boletus edulis* didactic model, composed of three fructifying bodies (mushrooms) in different stages of development. This model presents a rectangular base in wood painted black and varnished, Figure 1. In the bottom of the base, there is a label in German with detailed information about the species that occupies the entire area of the bottom surface. On the front, there is another label with a simplified identification of the mushroom and its edibility. The three mushrooms exemplifying the different stages of development are all papier-mâché sculptures made to a real-life scale, painted in colors similar to those of the represented species (white, brown and different shades of light brown) and fixed to the base. To simulate the habitat of this species of fungi (temperate woodland floors), real pleurocarpic moss species were glued to the base, around the mushroom stalk (“foot”), to reinforce realism. The cap (“hat”) of the “bigger” mushroom seems detachable from the body and the stalk is filled with a grey material, with a thin, vertical wooden structure inside. It is not known how the hat was originally placed, and both parts might have been produced separately and later joined, allowing a more didactic experience.

Although the sculptures seem structurally stable, the smaller mushroom has a crack at the bottom, though is still fixed to the base. Currently, the detached cap is not stable when placed on the stalk wood structure, and there is a risk of further damage if the object is not handled with care, Figure 2. This model also presents surface dirt and grime and some cracks in the chromatic layer, especially in the hats of the three mushrooms. The hat of the smaller mushroom also exhibits losses of the polychromy. The label glued to the front of the sculpture, besides some tears and losses, also exhibits an erosion of the surface of the paper possibly caused by insects belonging to the family Lepismatidae.

Inside the larger model mushroom hat, black spots were found, apparently caused by fungi. To evaluate this hypothesis, a sample was collected and observed under the microscope. The sample exhibited conidia and hyphae of a melanized filamentous fungus, with septate obpyriform conidia (apparently *Alternaria* sp.), which confirmed the presence of fungal colonization in the sculpture, Figure 3.

### 2.2. Analysis of the Papier-Mâché

#### 2.2.1. The Fibers under the Microscope

The paper pulp used in the papier-mâché technique applied in the studied model is mainly ground wood pulp. A few individual chemical pulp fibers were also observed in the papier-mâché sample collected from the hat of the largest mushroom, Appendix A. Wood pulps have also been previously identified in papier-mâché scientific models dating from the 19th century [10]. In ground wood pulps, all the wood components are retained, including lignin, which is responsible for the strong discoloration of this type of paper upon exposure to air and light [14]. Ground wood pulps have a high bulk and stiffness, but a low strength since lignin interferes with hydrogen bonding between fibers [14].

Both paper labels (front and bottom of the model) are short fibred and composed of chemical pulps. In chemical pulps, lignin is removed by chemical processes, which results in a higher quality paper, compared to ground wood paper. The front label also exhibits cereal straw in its composition [15], Appendix A. Straw fibers are shorter than the ones obtained from wood, and generally required mixing with these last ones for strengthening. Straw pulps were used in European papermaking from the 19th-century as a cheaper alternative to wood pulps [16].

#### 2.2.2. Molecular Characterization

The model with the higher stage of development was used to study the papier-mâché, as the analysis was possible inside this mushroom, Figure 2. Micro-XRF showed that the rod that was inserted inside the mushroom to attach it to the black base was made of wood. The main elements identified in the papier-mâché by micro-XRF were silicon (Si), potassium (K) and calcium (Ca); for more details see Appendix B and Appendix C. These elements could point to the use of a Si-O based material such as kaolin (Si_2_Al_2_O_5_(OH)_4_), which was unequivocally confirmed by its infrared spectrum, namely by its O-H stretching at 3700 cm^−1^, 3660 cm^−1^ and 3620 cm^−1^; its Al-O stretching at 960 cm^−1^; its O-H bending at 675 cm^−1^ and 630 cm^−1^, Figure 4. To allow for a good S/R spectrum for kaolin, the region where the Si-O stretching bands of kaolin as well as cellulose main bands absorb is saturated and therefore not shown. The presence of cellulose can be tentatively suggested by its O-H stretching at around 3348 cm^−1^ and the C-H stretching at 2902 cm^−1^. Infrared also detected the presence of a material similar to quartz (SiO_2_), Figure 4. The main bands, assigned to the Si-O stretching around 1085 cm^−1^, are again superimposed by the cellulose fingerprint, but the marker bands at 800 cm^−1^, 780 cm^−1^ and 695 cm^−1^ are clearly visible [13]. Importantly, carbon black was identified by Raman spectroscopy, which accounts for its greyish hue, Appendix C, Figure A4 (Appendix A).

In conclusion, the main components of papier-mâché are a quartz-based material and kaolin. A cellulose-based material is also present. The greyish color observed results from the use of carbon black. On the other hand, gypsum or any other form of calcium sulfate was not detected by infrared spectroscopy (this being a compound that would not be missed using this technique). So, plaster based on calcium sulfate was not used in this model.

### 2.3. Characterization of the Colors Used in the Boletus edulis

Micro-XRF was used to identify the main elements in the whites and two brown colors (light and dark brown) of the three fungi; in the whites, there was: zinc (Zn), barium (Ba) and sulfur (S); in the browns, there was: iron (Fe), silicon (Si), calcium (Ca) and potassium (K); for more details, see Appendix C. This first study shows that white and brown have two different compositions, and XRF data was used to obtain representative micro-samples of these colors.

The elements in the white suggest the presence of lithopone (ZnSBaSO_4_) or a mixture of zinc white (ZnO) and barium sulphate (BaSO_4_). With infrared spectroscopy, to the best of our knowledge, it is not possible to distinguish lithopone from barium sulfate. The infrared spectrum depicted in Figure 5, shows very clearly the presence of barium sulfate and calcium carbonate (CaCO_3_) by comparison with reference samples. The band at 1647 cm^−1^ together with the band at 3371 cm^−1^ suggests the presence of a proteinaceous binding media; being the first band attributed mainly to the C=O stretching of the amide (amide I band) and the second to N-H bending and C-N stretching (amide II band). Finally, Raman spectroscopy identified the white pigment as lithopone by its bands at 348 cm^−1^ and 988 cm^−1^ [17], Figure 6.

The presence of iron and silicon in the brown color detected by micro-XRF indicates the presence of an ocher-based material. However, it was not possible to identify the ocher using Raman spectroscopy. The infrared spectra show a heterogeneous sample in which the following compounds are detected: a material similar to quartz in a complex matrix and barium sulphate with a fingerprint similar to the white color. So it is possible that the pigment is also lithopone, in a proteinaceous binding media, Figure 7.

Part of the white color from the “bigger” porcini was covered with a varnish, Figure 8. This varnish was micro-sampled and its spectra revealed the fingerprint of a siccative oil; in Figure 8 it is compared with a linseed oil reference (unaged). It is possible that the lead observed in the XRF data is present in the varnish to catalyze the molecular drying of the oil.

In conclusion, a proteinaceous binding media was found for both white and brown colors. Lithopone is the pigment used for the white color and, possibly, it is also present in the brown. It was not possible to identify the pigment in the brown color, but the presence of iron indicates the use of an ochre.

### 2.4. Study of the Quadrangular Black Base

The three mushrooms are attached to a black quadrangular base in which, at the foot of the stem, is added an organic material, specifically, pleurocarpic moss. The base has two paper labels, one label with the identification of the species in Latin, and at the bottom of the base is another, in German, with the description of the mushroom. On this label is added an inventory seal and an institutional metal plate, Appendix B, Figure A3. The manuscript numbers on the inventory seal were identified by Raman as being made with carbon-based ink.

The paper used in the labels was discussed in Section 2.2.1. The metal plate is composed of copper (Cu) and zinc (Zn). The presence of Fe, Ca and Mg in the moss would agree with data collected from several mosses [18]. The moss is glued to the basis with a collagen-based glue, which is compared with parchment glue in Figure 9 [19,20,21]. 

In the black color used to paint the basis, the following main elements were detected: Fe and Ca. With infrared spectroscopy, it was possible to identify the binding media as a siccative oil and the varnish that covers it as a mixture of siccative oils with protein, Appendix C, Figure A8. It is one of the most complex colors studied, and there are possibly other compounds present that we have not been able to identify.

## 3. Materials and Methods

### 3.1. The University of Porto Herbarium and Visual Overview of the Condition of the 45 Models of Arnoldi’s Mushroom Collection

The University of Porto Herbarium (PO), in Portugal, was formally created in 1892, but the oldest specimens date back to 1863. In 1911, when the University of Porto was created, it was renamed as “Herbarium of the Faculty of Sciences” and later as “Herbarium of the Botany Institute Dr Gonçalo Sampaio”, in honor of the professor who expanded it, both in the number of specimens and in the typologies of collections. Since then, this Herbarium has remained linked to the University of Porto and is currently part of the biological collections of the Natural History and Science Museum of the University of Porto (MHNC-UP). Its collections include not only dried specimens (estimated in 130,000 vascular plants, bryophytes, lichens, fungi, and algae), of which ca. 300 correspond to types, representing mainly the Portuguese and European and African Floras, but also include specimens collected worldwide. The PO Herbarium collections comprise several ancillary specimens and objects such as seed and wood collections, microscopic slides, botanical fossils, field and laboratory notebooks, as well as ethnobotany, maps, scientific illustrations, and didactic models [3]. 

The general visual overview of the condition of the 45 models of Arnoldi’s mushroom collection housed at PO Herbarium was performed using a Dino-Lite Digital Microscope (Dino-Lite AM4113T), checking the structure and external materials observable at a maximum of 50x magnification. 

### 3.2. Points of Analysis for Boletus edulis Didactic Model and Micro-Samples for Infrared and Raman Analyses

The areas of analysis by microXRF and micro-sampling are described in Appendix B. 

Micro-sampling was performed with a micro-chisel from Ted Pella microtools under a Leica KL 1500 LCD microscope, (7.1× to 115× objective) and a Leica Digilux digital camera, with external illumination via optical fibers. Micro-samples were taken under a microscope, typically of 20–50 μm in diameter and as such invisible to the naked eye; as we have not yet obtained their weight, even though micro-scales have been used, we can use its detection limit to conclude that they weigh less than 0.1 μg.

### 3.3. Reference Samples Used in this Work

Kaolin, quartz, calcium carbonate, lithopone, and cellulose references are reagent grade chemicals. For more details see [13]. Collagen glue was prepared as described in [18].

### 3.4. Energy-Dispersive X-ray Fluorescence Spectroscopy (microXRF) 

X-ray fluorescence data were obtained with an ArtTAX spectrometer of Intax GmbH, with a molybdenum (Mo) anode, a Peltier cooled SDD (Silicon Drift Detector), model Xflash, installed in a mobile arm of the spectrometer, was used to collect the produced X-rays. The use of pollycapillary lens allows beam spot analysis with a spatial resolution of 70 µm. The experimental parameters used were: 40 kV of voltage, 300 µA of current intensity, and 150 s of acquisition time.

### 3.5. Micro-Fourier Transform Infrared Spectroscopy

Infrared analyses were performed using a Nicolet Nexus spectrophotometer coupled to a Continu*μ*m microscope (15× objective) with an MCT-A detector. The spectra were collected in transmission mode, in 50 µm^2^ areas, resolution 4 or 8 cm^−1^ and 128 scans, using a Thermo diamond anvil compression cell. CO_2_ absorption at ca 2400–2300 cm^−1^ was removed from the acquired spectra (4000–650 cm^−1^). To improve the robustness of the results, at least two spectra were acquired from different sample spots. 

### 3.6. Raman Spectroscopy

Raman microscopy was carried out using a Horiba Jobin Yvon LabRAM 300 spectrometer, equipped with a diode laser with an excitation wavelength of 785 nm and a maximum laser power of 37 mW measured at the sample. Spectra were recorded as an extended scan. The laser beam was focused with a 50× Olympus objective lens and the spot size was of 4 μm. The laser power at the sample surface was between 9.5 and 0.37 mW. No evidence of degradation was observed during spectra acquisition. More than three spectra were collected from the same sample. A silicon reference was used to calibrate the instrument.

Handheld Raman spectroscopy was carried out with a Raman Mira DS, equipped with a laser emitting light at 785 nm with a maximum power of 100 mW, within a spectral range of 200–2300 nm. This equipment provides a spectral resolution of 8–10 cm^−1^ and features a measuring spot of 0.042–2.5 mm. Signal acquisition is performed using Orbital Raster Scan (ORS) and involves averaging the signal collected from relatively large sample areas while maintaining the desired resolution. All spectra were acquired with the maximum laser power and averages, varying the integration time according to the target material and working distance (the higher the distance, the higher the acquisition time). A minimum of three spectra were collected from the same sample to ensure reproducibility.

### 3.7. Characterization of Paper Pulp Types

Samples of paper fibers were placed on a glass slide in a water drop, separated with probes and dried over a heating plate for further analysis. Paper pulp type was determined using Herzberg reagent and following TAPPI 401 standard (TAPPI, 2008) for stain solution preparation [22], sample analysis procedure and interpretation of results. The morphology and coloration of the fibers after reaction with the Herzberg solution were analyzed using an Axioplan 2ie Zeiss optical microscope under a bright field and cross polarization, and recorded with a digital Nikon camera DXM1200F. 

### 3.8. Analysis of the Presence of Fungi

A sample of a varnish layer showing black stains was collected from the inside of the model mushroom hat with a micro-chisel from Ted Pella microtools under a Leica KL 1500 LCD microscope and placed on a glass slide. A drop of lactophenol blue solution was added and a cover glass was placed on top. The sample was analyzed using an Axioplan 2ie Zeiss optical microscope under a bright field, and recorded with a digital Nikon camera DXM1200F.

## 4. Conclusions

We consider that the current study of the *Boletus edulis* model was pivotal in launching the overview of MHNC-UP collection’s importance in the context of university museum collections and Arnoldi’s heritage in the world. With this first approach, we collected data in order to proceed with the condition evaluation of the remaining models of the collection and plan a conservation strategy for these objects that have outlived their didactic intensive use for more than a century.

The study of materials and techniques used in the construction of the didactic model of *Boletus edulis* was initially planned based on techniques that do not require micro-sampling. However, one of the important techniques necessary to accurately identify inorganic compounds, handheld Raman, did not give positive results in most cases, due to the fluorescence of binders and varnishes. For this reason, micro-samples of all components were acquired, which were analyzed using microRaman and microFTIR. These micro-samples are invisible to the naked eye, and as such, it is not possible to mount them as cross-sections for observation under a microscope (or SEM-EDS).

The multi-analytical approach used allowed accurate identification of most of the components. The papier-mâché is prepared with ground wood pulp, kaolin and a quartz-based material. Its grey color is due to the presence of the pigment carbon black. The white paint uses lithopone in a proteinaceous tempera, with calcium carbonate as filler. A proteinaceous binding media was also detected in the brown color together with a quartz-based compound, barium sulphate with a fingerprint similar to the white color, so it is possible that here also the pigment used is lithopone. In this brown, iron was identified by microXRF, pointing to the use of ocher. The moss was fixed to the rectangular base using a collagen-based glue. This rectangular base was painted in black and varnished, and the infrared spectrum obtained of both color paint and varnish was one of the more complex obtained. In the black and in the varnish we identified a drying oil; other components can be present that we were unable to identify. In the study by Manso et al., a zinc-based white is also used and authors have identified the brown color likely as umber [7]. 

In Mayoni’s study of papier-mâché plant models [10], the following stratigraphy was proposed for the construction of color: a first layer based on calcium carbonate (although the SEM-EDS analyzes also identify the element Al and Si); a second white layer in which Zn was detected as “very pure”; a third layer in which Zn and Pb were identified. The first observed layer may be related to the composition we identified in *B. edulis* papier-mâché; in the second and third layers, the presence of only Zn could indicate the use of another white (for example, zinc white); the presence of Pb in the third layer may be associated with the varnish. The varnish we identified is based on a drying oil, and Pb may be there as PbO to accelerate drying.

This is the first study that allows an in-depth and accurate identification of the materials and techniques used in the production of the 19th-century didactic model of *Boletus edulis* from the MHNC-UP Herbarium. These complex heritage materials are also built in a multilayer stratigraphy in which different binding media and varnishes were used. We consider that the modus-operandi tested here can now be applied to a larger selection of the Arnoldi collection of mushroom models. This information is useful to better understand the deterioration mechanisms observed on this kind of object. In the future, this information will also be important for selecting the most appropriate materials for conservation, possibly within a funded research project.

## Figures and Tables

**Figure 1 molecules-28-01062-f001:**
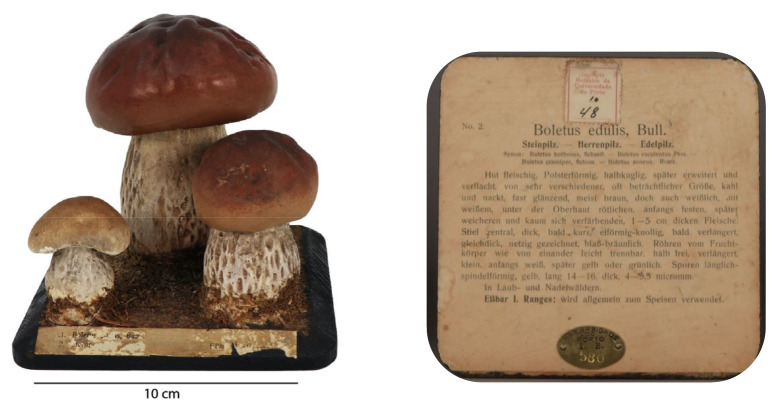
Model *Boletus edulis,* Bull. (**left** image) and label glued on the base (**right** image).

**Figure 2 molecules-28-01062-f002:**
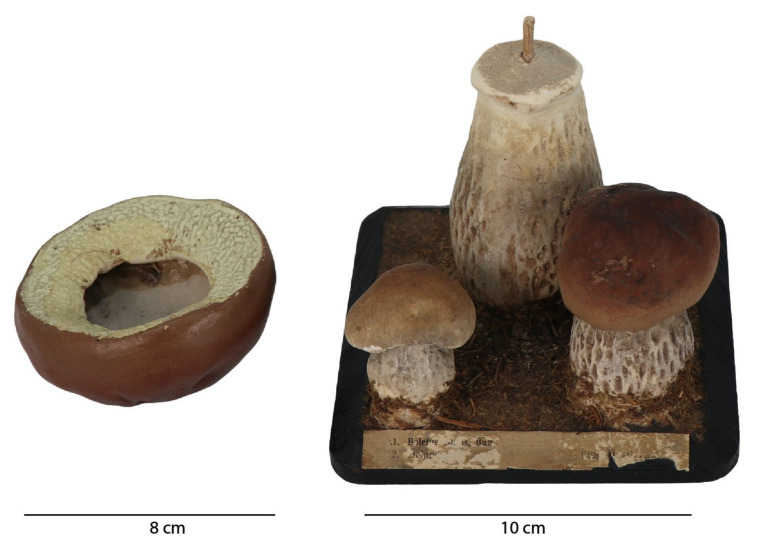
Model *Boletus edulis,* Bull. model (**right**) and the detached cap (**left**).

**Figure 3 molecules-28-01062-f003:**
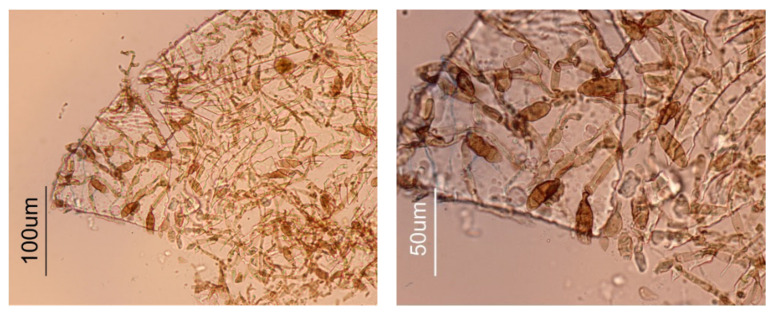
Optical microscope images (brightfield) of a sample collected from the inside of the mushroom hat, at lower (**left**) and higher (**right**) magnification.

**Figure 4 molecules-28-01062-f004:**
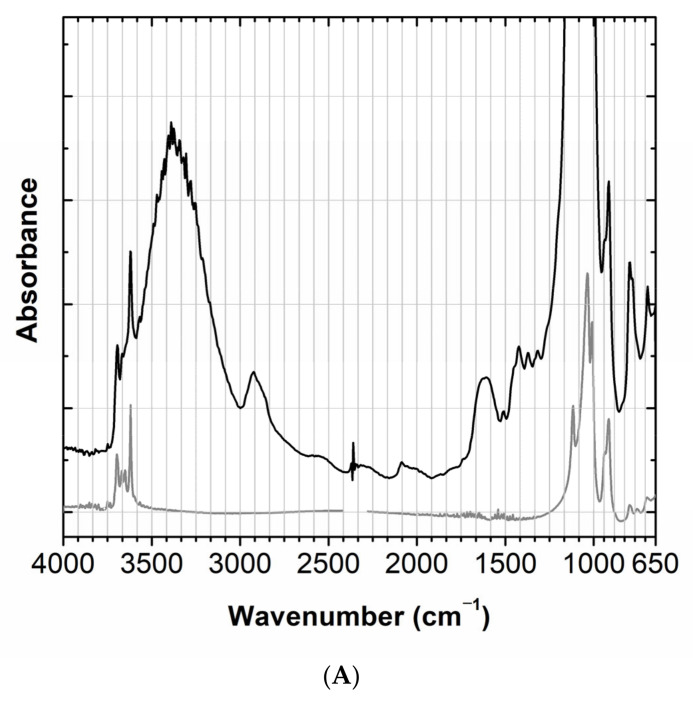
(**A**) Infrared spectra of papier-mâché (black) compared with a kaolin reference (grey)**.** (**B**) Infrared spectrum of papier-mâché (black) compared with a quartz reference (grey), in the 1500–650 cm^−1^ region; full spectra in Appendix C, Figure A1.

**Figure 5 molecules-28-01062-f005:**
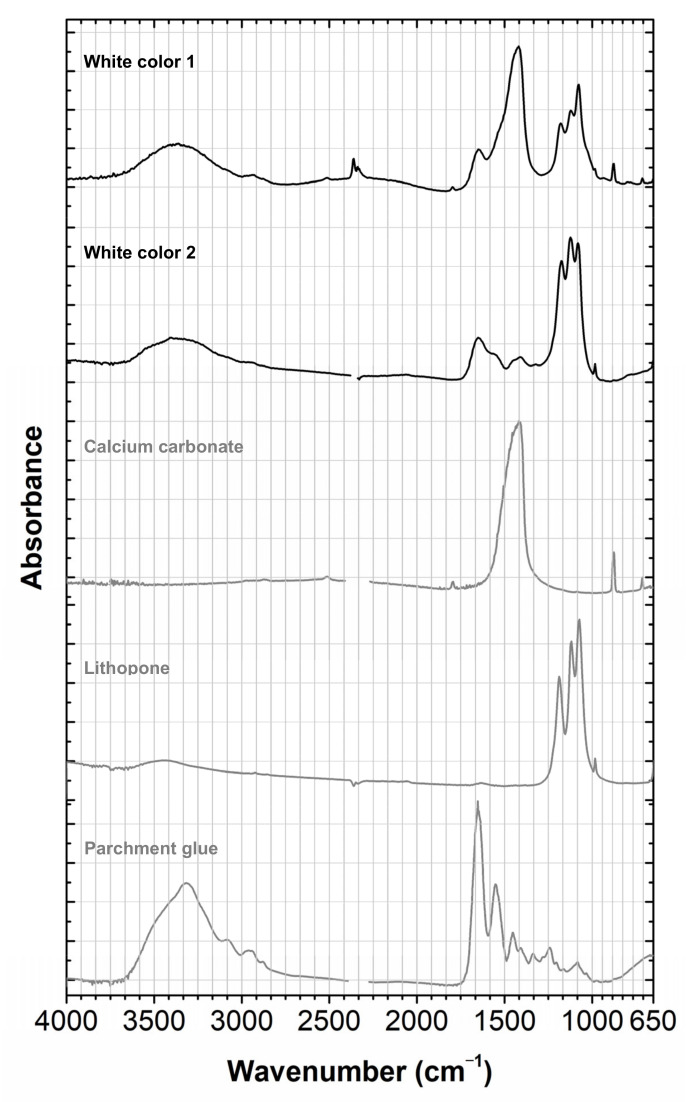
Infrared spectrum of the white color compared with selected references. The white color is a complex formulation, in which the white pigment was identified as lithopone by microRaman in a proteinaceous media, having calcium carbonate as a filler. For more details on white 1 and 2, see Appendix B.

**Figure 6 molecules-28-01062-f006:**
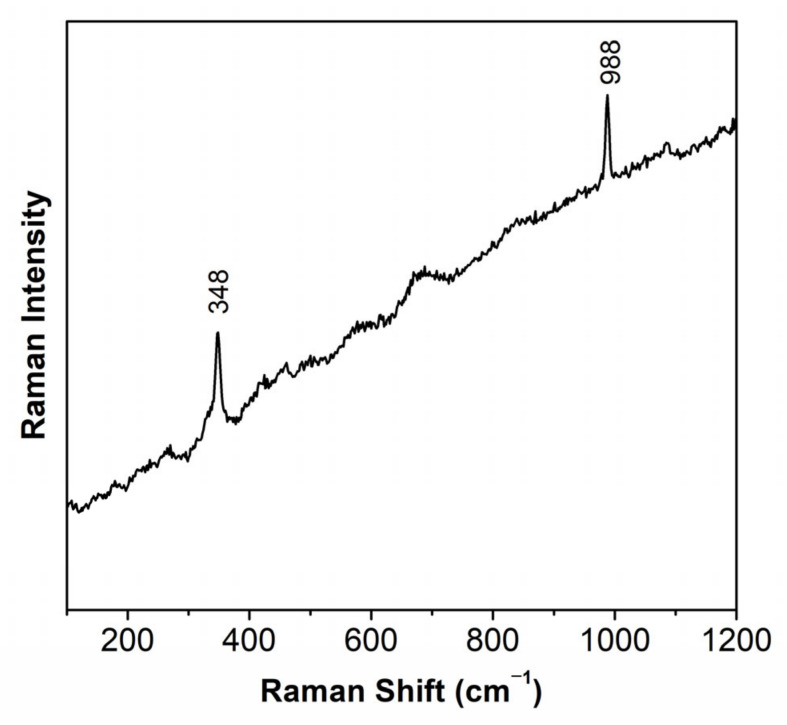
Raman spectra of the white color, identifying the presence of lithopone.

**Figure 7 molecules-28-01062-f007:**
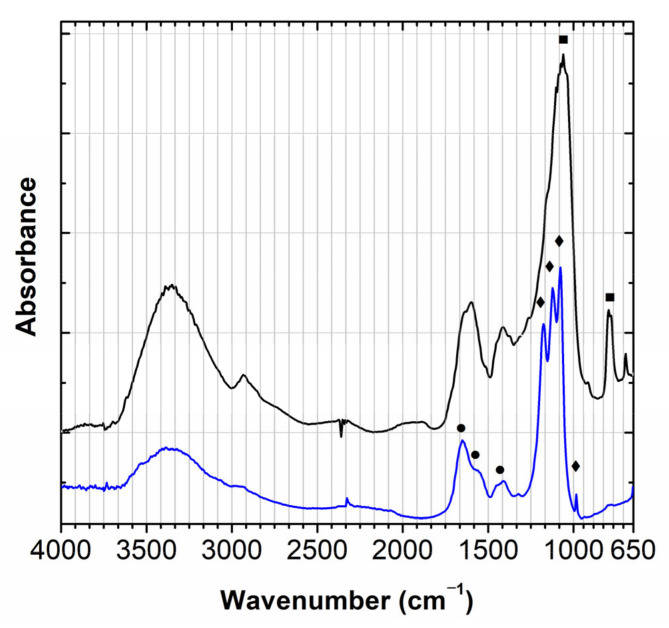
Infrared spectra of brown colors (black and blue), in which the main bands detected are assigned to quartz (■), barium sulfate (◆) and a proteinaceous binder (●).

**Figure 8 molecules-28-01062-f008:**
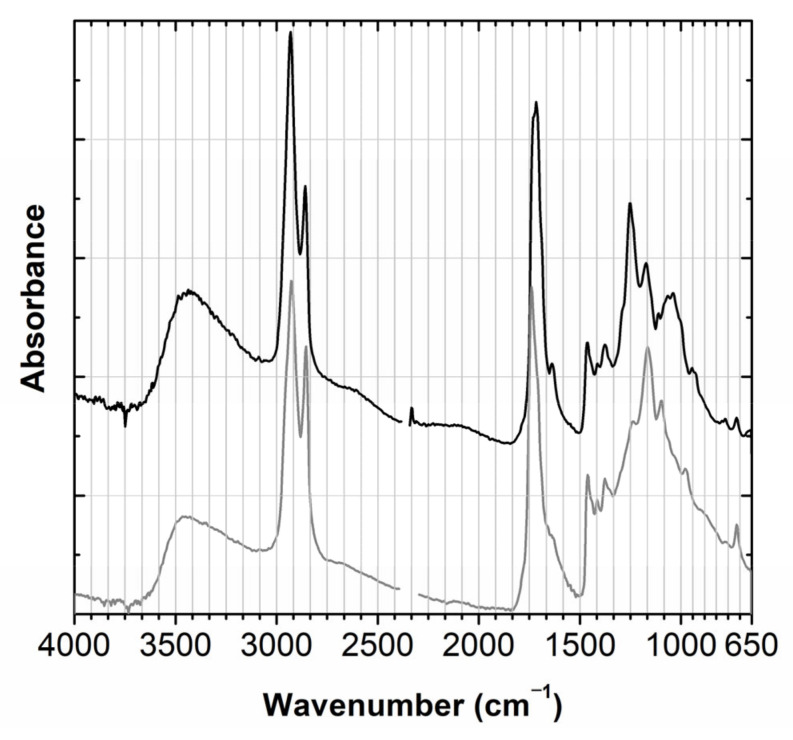
Infrared spectrum of the varnish applied over the white color (black) compared with a linseed oil reference (grey).

**Figure 9 molecules-28-01062-f009:**
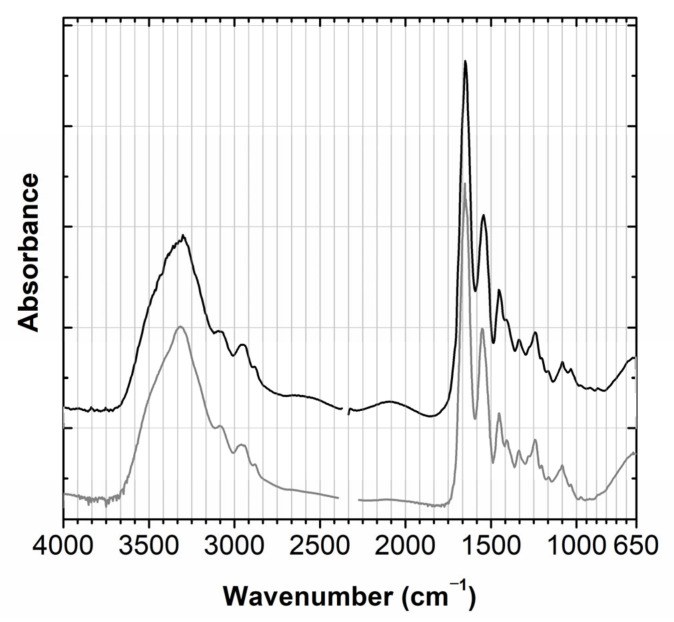
Infrared spectrum of the glue used to fix the moss to the base (black) compared with a parchment glue reference (grey).

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
