# Peer review of "Study of the Materials and Techniques of a Rare Papier-Mâché Mushroom Model Crafted in H. Arnoldi Factory"

_molecules, 2023, doi:10.3390/molecules28031062_

Round 1
Reviewer 1 Report
Dec 29, 2022 Review
General Comments. The article is reporting on the analysis of botanical models, paper mache and paint used for coloring. Also, fungi infestation was detected and briefly reported. According to the statement in the Abstract, the motivation for undertaking the extensive analysis was to assist conservation of the models. However, there was no evidence that the analysis was used in any conservation procedure, nor an indication of any conservation is actually planned in the future. Was it really needed? If so, how the analysis will assist a conservator? If that was the intention behind undertaking the analysis, its future application needs to be clarified.
The sections need to be rearranged; Materials and Methods need to be introduced before Results and Discussion. Line comments address some of the suggested changes. Not all statements regarding paper pulp are accurate, physical properties are misinterpreted and their correlation with molecular composition is oversimplified. Other statements are broad and not substantiated as to how the conclusions were reached.
The description of the models belongs to the Materials section. How many of the mushroom models were actually examined? That was never specified.
Some of the sentences need to be clarified.
Line comments.
Title: either all nouns start with a capital letter or none; needs consistency.
Abstract: The authors justify the motivation of undertaking such extensive analysis of paper mache and paint pigments as fundamental for carrying out conservation studies. It is not clear if the authors planned to replicate the use of the original material, or just acquire knowledge of the materials that were used in making the models.
17…houses a collection, rather than preserves
19 how many of these models were examined… all?
20 plural…models
27..past tense..was prepared
53 is there a reason why ‘300 Types’ is written in the capital letter?
54 exsiccates… probably replacing with ‘dried specimens’ would simplify. ‘Exsiccate’ is typically used as a transitive verb rather than a noun. What is meant by ‘national’- which country is referred to?
51-55 This sentence needs to be rewritten. It starts with ‘dry specimens, and next mentions ‘exsiccates, which are also dried objects…
61 difference
64 in what country, city, not in Porto…
143 …were wooden forms baked too? That’s what the sentence is implying. This sentence needs clarity.
155.. described- where? Is that a reference to a catalog or an early record?
177 ..there is no mention as to how any of the analysis was carried out, no parameters of any of the analysis, especially spectroscopy, no magnification or type of microscopes used, no pressure or magnification in SEM-EDS, etc. How many of the 45 specimens were examined? All or some? That needs to be elaborated.
179.. what components were analyzed? Was it an overall examination?
186 the bibliographic reference leads to a paper on yellow chromium. What is its relation to this article?
Section 1.3 change to past tense.
190 2.1 remove ‘conservation’ … it is about condition..
Half of section 2.1 belongs to the description of materials, or ‘Samples’ that underwent testing, rather than to “Results”. There is no section referring to ‘Materials’ and/or structure and composition of the models. That needs to be separated into one section.
209 only one specimen showed structural damage or only one was examined?
223 restructure the sentence.
225 fungi are plural of fungus…was only one type identified, or were other types of filamentous fungi?
227 fig 3 appears to be Alternaria solani that often produces dark fruiting structures. Are they melanized? that is not certain.
2.2.1 How the chemical and physical characterizations were derived? Authors seem to be mixing molecular characteristics of lignin with physical attributes. A small percentage of ‘chemically’ processed pulp…how the percentage was determined? Is straw pulp really of good quality?
242 straw is a generic name and encompasses many different plant residues from crops.… what straw was identified in this paper mache? Lignin content also varies between plant species from which straw was obtained.
2.2.2 vibrations listed here do not necessarily correspond with common kaolin characteristics. For example, other authors use Si-O vibration as ‘finger print’, pointing to 1032 cm-1 and 1010 cm-1 or 1633 cm-1 for on -OH, etc, in an article “Characterization of technical kaolin by XRF, SEM, XRD and FTIR.. by Dewi et al, 2018….The findings listed by the authors need to be presented in context and discussed in light of other available references. More specifics as to how the analysis was carried out are needed to justify the conclusions.
272 Section 2.3 what were the conditions of XRF analysis?
282 what were the reference samples? Was it mentioned earlier?
289 the presence of Fe or Si can be determined with other instruments not necessarily Raman.
335 ‘Materials and Methods’ section needs to be inserted before the Discussion.
340 Sources of reference samples need to be identified- are they laboratory grade? All relevant information needs to be included in the text rather than referring the reader to Appendix.
372 what was the Raman laser power, at the sample and at sources? What was the time and intensity of exposure?
389 Conclusion
How has this extensive analysis assisted conservation? Was any conservation of the mushroom models undertaken using the outcome of the analysis?
Author Response
Please see pdf file

Reviewer 2 Report
The study might have some potential impact, but I think it needs major revision before it can be accepted.
Main issue #1: strengthen the novelty and importance of this particular case study, as there are numerous papers in the literature using similar analytical setups to investigate multicomposite works of art.
Suggestions: intro: move closer to the intro start the part with the importance of the Porto collection as compared to other herbaria; this underpins the rationale for the study.
Discussion: the complementarity of Raman and µFTIR should be better stressed, highlighting cases where one technique complements the other.
Main issue#2: the authors state that “this new knowledge is fundamental to informing the choice of the best conservation strategies for the preservation of these extraordinary models.”. This is correct, but I see this type of sentence in numerous papers where no practical link is outlined between the analytical study and possible materials/methodologies for the intervention. While detailing the intervention is out of the scopes here, at least a paragraph at the end of the discussion should be added to show how the analytical results can be useful in providing potential suggestions to restoration methods. The authors can find some fundamental references to current state-of-the-art advanced restoration materials/methods to include in this discussion to show the potential use of their analytical study.
Other issues:
Figure 4: please include a legend to materials names in the figure boxes (e.g. “kaolin reference” close to its spectra, etc.)
Figure 6: please check the legend: n, t, l, do not correspond to symbols use in the box, please fix.
In general, the format of the FTIR figures could be improved and adapted to current standards in journals, even though this is only aesthetical preference.
Finally, technical figures in the main text only cover FTIR results, I suggest re-editing to distribute contents among all the used techniques, and moving some of the FTIR data to the SI.
Author Response
Please see pdf file

Round 2
Reviewer 2 Report
the Authors have addressed some of the main points and improved the manuscript quality
Author Response
Reviewer2's comments and suggestions were: "The Authors have addressed some of the main points and improved the manuscript quality". Therefore, we think the reviewer recommends the publication of our manuscript.
English language and style were considered fine/minor spell check required. So we have performed a minor spell check.